# Development of an asthma policy model for Canada: Lifetime Exposures and Asthma outcomes Projection

Tae Yoon Lee[1,2]*, John Petkau[3], Kate M. Johnson[2,4], Stuart E. Turvey[5], Amin Adibi[2], Padmaja Subbarao[6], Ainsleigh Hill[2], Mark Ewert[2], Mohsen Sadatsafavi[2]

1 Quantitative Sciences Unit, Stanford School of Medicine, Stanford, California, United States of America, 2 Respiratory Evaluation Sciences Program, Faculty of Pharmaceutical Sciences, University of British Columbia, Vancouver, British Columbia, Canada, 3 Department of Statistics, University of British Columbia, Vancouver, British Columbia, Canada, 4 Division of Respiratory Medicine, Department of Medicine, University of British Columbia, Vancouver, British Columbia, Canada, 5 Department of Pediatrics, BC Children's Hospital and The University of British Columbia, Vancouver, British Columbia, Canada, 6 Division of Respiratory Medicine and Translational Medicine, Department of Paediatrics, Research Institute, The Hospital for Sick Children, Toronto, Ontario, Canada

* harrytyl@stanford.edu

## Abstract

### Objectives

To address the need for health policy planning focused on early interventions for asthma, we developed the Lifetime Exposures and Asthma outcomes Projection (LEAP), a reference policy model for asthma in Canada.

### Methods

Based on a previously developed concept map, we used an open-population micro-simulation design to deal with the multidimensionality of risk factors and the need for modelling realistic aspects of adopting health technologies. The model consists of five intertwined modules: demographics, risk factors, asthma occurrence, asthma outcomes, and payoffs. The demographics module was based on sex- and age-specific estimates and projections from national surveys. For the first version of the model, we concentrated on key risk factors from the concept map: age, sex, family history of asthma at birth, and exposure to antibiotics in the first year of life. The distributions of risk factors were estimated from population-based administrative databases and a population-based longitudinal birth cohort. Quantitative evidence synthesis provided estimated parameters for the asthma occurrence and outcomes modules. Costs and utility weights were obtained from the literature. We conducted face validity, calibration, and internal validity assessments.

**Data availability statement:** The datasets generated and/or analysed during the current study are available in the GitHub repository (https://github.com/tyhlee/LEAP.jl). Restricted datasets that support the findings of this study are the Chronic Disease Registry (https://assets-hdpbc.healthbc.org/doc/show/hdp/d06baaea-6692-4118-90ca-7de0a85177ea) from the Ministry Health of British Columbia (complete the following instructions here to gain access: https://www2.gov.bc.ca/assets/gov/health/forms/5456fil.pdf) and the Canadian Healthy Infant Longitudinal Development Study Data (https://childcohort.ca/for-researchers/data-access/), but restrictions apply to the availability of these data, which were used under license for the current study, and so are not publicly available. Please refer to the instructions in the URLs provided to obtain access to the restricted datasets.

**Funding:** This study was funded by a research grant from Genome Canada (274CHI). The funders had no role in any aspect of this study and were not aware of the results.

**Competing interests:** The authors have declared that no competing interests exist.

## Results

LEAP is capable of modelling asthma-related health outcomes at the individual level from 2001 onwards. Age-sex stratified demographic projections from the model closely matched projections, while asthma prevalence and control levels matched their respective estimates from the administrative data and estimates from the literature. The LEAP model is publicly accessible as open-source software: https://github.com/tyhlee/LEAP.jl (Julia) and https://resplab.github.io/leap/ (Python).

## Conclusions

LEAP is the first reference Canadian asthma policy model that emerged from identified needs for health policy planning for early interventions in asthma. It can provide a unified framework under which different interventions and policies can be consistently compared to identify those with the highest value proposition. The model needs to be gradually updated to accommodate other risk factors, and its validity should be independently examined.

## 1. Introduction

Asthma is an early-onset, chronic disease of the airways affecting over 3 million Canadians and 260 million people worldwide [1,2]. By all accounts, asthma imposes a substantial economic and humanistic burden on individuals, their families, and society. It is one of the leading causes of emergency department visits, hospital admissions, missed school days for children, and loss of work productivity for adults [3–5].

While there is no definite cure for asthma, emerging evidence promises the development of innovative interventions that might reduce the risk of asthma onset in children. For example, researchers have found an association between exposure to antibiotics during infancy and the risk of asthma, a relationship that seems to be mediated by the gut microbiome [6,7]. Thus, interventions and policies that reduce unnecessary exposure to antibiotics may prevent the development of asthma among children [8].

Efficient health policymaking relies on projections of future consequences of decisions made today. Hence, epidemiological forecasting, burden of disease projections, and economic evaluations are foundational to evidence-informed policymaking. Stakeholders must know the value of competing interventions to make informed decisions. For the most part, projecting health outcomes at the population level requires creating a decision-analytic model of the disease of interest and quantifying the impact of interventions under evaluation [9]. Such decision-analytic models have been used for translating evidence from clinical trials into practice, improving delivery of care, and informing policy and clinical guidelines [10]. Examples include models for lung cancer [11] and Type 2 diabetes [12].

Customarily, a new decision-analytic model is developed to conduct policy analysis and economic evaluation for each set of new health interventions. However, this

approach has been criticised for several reasons [13]. Such 'piecemeal' modelling is inefficient, because different models are sometimes developed and used for the same disease, resulting in waste of resources. Additionally, this approach can lead to inconsistency in model structure, evidence synthesis, and underlying assumptions. An alternative approach to address these issues is the use of a 'reference' model that serves as a unified framework for evaluating different interventions for the same disease [13]. The reference model must be transparent enough so that users can understand its structure and assumptions and use it with confidence.

A recent scoping review and an earlier systematic review concluded that current decision-analytic asthma models generally do not consider the multifaceted and heterogeneous nature of the disease, lack transparency, lack sufficient granularity to model the nuances of interventions (e.g., imperfect adherence to treatments), and do not fully adhere to best practice recommendations on modelling [14,15]. The absence of high-quality modelling platforms hinders the implementation of interventions that could substantially reduce the burden of asthma. Recognising this knowledge gap, key stakeholders in Canada have called for a national reference policy model for asthma [14]. The overarching goal of this study was to develop a reference policy model for evaluating interventions for asthma in Canada.

## 2. Methods

This study was approved by the institutional review board of the University of British Columbia, Vancouver (H22-00571). Data analysis and evidence synthesis were conducted during the period between June 1, 2022 and June 30, 2025. We did not have access to information that could identify individual participants.

Our model development underwent the following stages, in accordance with the best standards [16–19]. Based on a previously developed concept map [14], we selected an appropriate model structure and identified key model components. We performed several analyses to generate and synthesise evidence required for the model and calibrated the model to the Canadian population. We then implemented the model as open-source, open-access software. Lastly, we carried out face validity and internal validity assessments to evaluate the consistency of evidence synthesis, underlying assumptions, and model implementation.

The process started with a concept map, developed by deep engagement with a steering committee of economic modellers, allergists, and respirologists across Canada [14]. The committee's focus was on childhood asthma given that most preventive strategies pertain to this group. The model concept, which emerged through a modified Delphi process [14], includes three major groups of risk factors related to diagnosis of childhood asthma: patient characteristics, family history, and environmental factors. Further, the steering committee recommended simulating individuals (rather than the population as a whole) to deal with the multidimensionality of risk factors and capture the complex interactions among decisions. They also recommended an open-population structure to model realistic aspects of implementing an intervention (e.g., gradual market penetration of health technologies). An open-source, open-access platform was deemed necessary to make the model accessible to the wider research community and to allow for independent evaluations and validations.

Our implementation of this conceptual model is as an open-population discrete-time microsimulation, in which a virtual individual is created to represent each person in the population of interest. In each time cycle, actions and behaviours of the individual are simulated based on pre-defined rules, and the attributes and disease characteristics of the individual are updated accordingly. We set the time cycle unit to annual as we concluded that an annual time cycle strikes a right balance between granularity in simulation events over time and the computational demands of the planned analyses.

The model is open-population as it simulates the entire Canadian population over time, including birth, death, immigration, and emigration. The time horizon is defined in calendar time. The maximum range of the time horizon is from 2001 (the year subsequent to the earliest year for which we had access to data to examine asthma outcome trends) to 2065 (the latest year for which population projections are available), enabling both retrospective and prospective simulations. If the base year is set to a time after 2019, the latest year for which data on population structure is available, the model internally will simulate the 2019 population towards the base year.

## 2.1. Model structure and components

The model consists of five intertwined modules: 1) demographics, 2) risk factors, 3) asthma occurrence, 4) asthma outcomes, and 5) payoffs. The core of the model can be conceptualised as a series of stochastic structural equations relating risk factors to each other and to the events of interest (**Fig 1**). **Table 1** summarises the structural equations in the asthma model by modules. Parameter values are provided in a separate table online (https://github.com/tyhlee/LEAP.jl).

In the following subsections, we describe each of the five modules in detail and then provide an overview of how these modules are connected including pseudocode for the microsimulation model at the end.

## 2.2. Demographics module

The demographics module consists of birth, immigration, emigration and mortality equations. At the start of the simulation, an initial population is generated for the specified base year. In subsequent years, virtual individuals enter the simulated population through birth or immigration according to the estimates or projections of population growth and aging, and exit when one of the following events occur: death, emigration, or reaching the end of the time horizon.

To model the initial population prior and up to 2019, we used the population estimates by sex and age from Statistics Canada [20]. To model the population growth and aging for later years, we adopted population projections by sex and age from Statistics Canada [21]. Details of the demographics module are provided in Supplementary Materials Section 1 in S1 File.

## 2.3. Risk factors module

For the first version of the model, guided by consultation with the steering committee, we concentrated on several risk factors from the concept map: age, sex, family history of asthma at birth, and infant (<1 year old) exposure to antibiotics. This represents at least one risk factor from each risk factor group in the concept map [14]. Age and sex of a virtual individual were determined by the demographics module. To estimate the probability of having a family history of asthma at birth,

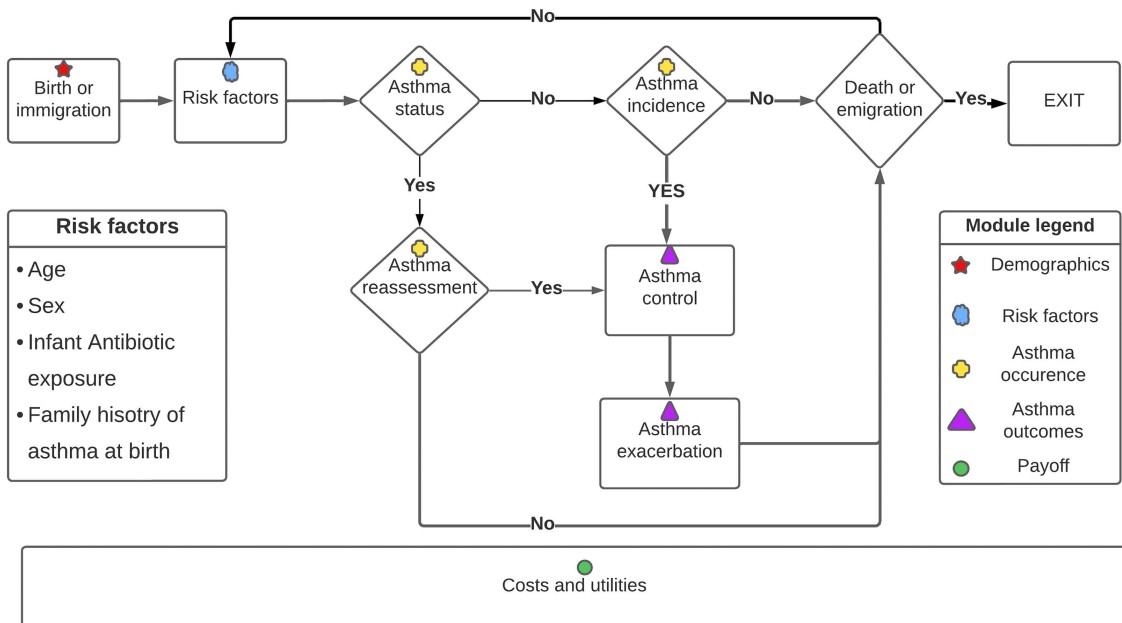

**Fig 1. Schematic illustration of the reference asthma policy model.**

**Table 1. Structural equations by modules in the asthma model.**

| Modules | Description | Source/Reference |
|---|---|---|
| **Demographics** | | |
| Initial population | For a given year $\in \{2001, \ldots, 2019\}$, the Statistics Canada estimated population size is used. The empirical distribution of sex and age for the given year is used to generate age and sex of each individual $i$ in the initial population:<br>$(sex_i, age_i) \sim Categorical\left(p_{(sex, age)}|year\right)$. | Statistics Canada [20] |
| Birth | For a given year and projection scenario, the Statistics Canada estimated or projected number of births is used. The empirical or projected distribution of sex at birth for the given year is used to generate sex of each birth $i$ (other risk factors are described under the Risk factors module below):<br>$sex_i \sim Bernoulli\left(p_{sex}|year, projection\ scenario\right)$. | Statistics Canada [20,21] |
| Net immigration | For a given year and projection scenario, the number of net immigrants by sex and age is estimated by calibration. The empirical or projected distribution by sex and age is used to generate sex and age of each net immigrant $i$:<br>$(sex_i, age_i) \sim Categorical\left(p_{(sex, age)}|year\right)$. | Calibration |
| Net emigration | For a given year and projection scenario, the number of net emigrants by sex and age is estimated by calibration. The empirical or projected proportion of net emigrants by sex and age is used to generate whether individual $i$ emigrates:<br>$emigration_i \sim Bernoulli\left(p|sex, age, year, projection\ scenario\right)$. | Calibration |
| Mortality | For a given year, sex, and age, the Statistics Canada corresponding estimated or projected life table is used to estimate the probability of death $p$ and generate whether individual $i$ dies:<br>$death_i \sim Bernoulli\left(p|year, sex, age\right)$. | Statistics Canada [22] |
| **Selected risk factors[b]** | | |
| Family history of asthma at birth (FHA) | Regardless of birth year, sex, and other characteristics, $FHA_i$ (the indicator of having a parental history of asthma at birth) for individual $i$ is generated:<br>$FHA_i \sim Bernoulli(p)$. | CHILD study [23] |
| Infant antibiotic exposure (IAE)[a] | For a given year and sex, $IAE_i$ (the number of courses of antibiotics individual $i$ receives in the first year of life) is generated by a negative binomial model. To prevent unrealistic extrapolation, the mean parameter is truncated at 0.05 (equivalent to 50 per 1,000 infants).<br>$IAE_i\ (sex, year) \sim NegativeBinomial\left(\max(0.05, \mu), \theta\right)$;<br>$\log(\mu|sex, year) = \beta_0 + \beta_1 sex + \beta_2 \mathbf{1}\left[year > 2005\right] + \beta_3 year + \beta_4 \mathbf{1}\left[year > 2005\right] year$. | British Columbia Ministry of Health [24] |
| **Asthma occurrence** | | |
| Effect of family history of asthma (FHA) on asthma prevalence (prev) and incidence (inc) | The effect of FHA in terms of log odds ratio (OR), denoted as $FHA_j(FHA, age)$, on $j \in \{inc,\ prev\}$ is given by the following equation:<br>$FHA_j(FHA, age) := \log\left(OR_j|FHA, age\right)$<br>$= \mathbf{1}\left[age \geq 3\ AND\ FHA = 1\right]\left(\beta_{0j} + \beta_{1j}\left(\min(age, 5)\right) - 3\right)$. | Patrick et al. [7] |
| Effect of infant antibiotic exposure (IAE) on asthma prevalence and incidence | The effect of IAE in terms of log OR, denoted as $IAE_j(IAE, age)$, on $j \in \{inc,\ prev\}$ is given by the following equation:<br>$IAE_j(IAE, age) := \log\left(OR_j|IAE, age\right)$<br>$= \mathbf{1}\left[(age \geq 3\ AND\ age \leq 7)\ AND\ IAE > 0\right]$<br>$\left(\beta_{0j} + \beta_{1j}(age - 3) + \beta_{2j}\min(IAE, 3)\right)$. | Lee et al. [8] |
| Asthma prevalence[c,d,e] | For a given year and individual characteristics, $Y_{prev}$ (the indicator of having an asthma label) is generated:<br>$Y_{prev,i} \sim Bernoulli\left(p|year, sex, age, FHA, IAE\right)$;<br>$logit\left(p|year, sex, age, FHA, IAE\right) = \beta_0 + \vec{\beta}_1 poly(year, 2) * sex * poly(age, 5) + FHA_{prev}(FHA,\ age) + IAE_{prev}(IAE, age)$<br>Note that the length of $\vec{\beta_1}$ is 34. For each of males and females, there are 2 main effects of year and $year^2$, 5 main effects of age, $age^2$, $\ldots$, $age^5$ and the corresponding 10 (first order) interaction effects between year and age. | British Columbia Ministry of Health [24], calibration |
| Asthma incidence | For a given year and individual characteristics, $Y_{inc}$ (the indicator of developing an asthma label) is generated:<br>$Y_{inc,i} \sim Bernoulli\left(p|year, sex, age, FHA, IAE\right)$;<br>$logit\left(p|year, sex, age, FHA, IAE\right) = \beta_0 + \beta_1 year + \vec{\beta_2} sex * poly(age, 5) + FHA_{inc}(FHA,\ age) + IAE_{inc}(IAE,\ age)$<br>Note that the length of $\vec{\beta_2}$ is 10. For each of males and females, there are 5 main effects of year, $year^2$, $\ldots$, $year^5$. | BC Ministry of Health [24], calibration |

*(Continued)*

| Modules | Description | Source/Reference |
|---|---|---|
| Asthma reassessment | For a given year, sex, and age, an individual labelled as having asthma is reassessed. They maintain their asthma label with probability $p$, where $p$ is determined by calibration:<br>$\text{asthma reassessment}_i \sim \text{Bernoulli}\left(p\|\text{year, sex, age}\right)$. | Calibration |
| **Asthma outcomes** | | |
| Asthma control | There are three levels of asthma control: 1 = uncontrolled (UC), 2 = partially controlled (PC), and 3 = well-controlled (WC). For individual $i$ labeled as having asthma, the proportion of time spent in each of the three asthma control levels in a year (given their age and sex) is given by the probabilities generated by a random-effects ordinal regression model. For the label of the asthma control level $Y_i$:<br>$logit\left(P\left(Y_i \leq j\|\text{sex, age}\right)\right) = \theta_j - \left(\beta_{0i} + \overrightarrow{\beta}_1 \text{sex} * \text{poly}(\text{age}, 2)\right)$, for $j = 1, 2$, where $\beta_{0i} \sim N\left(0, \sigma^2\right)$<br>Note that the length of $\overrightarrow{\beta}_1$ is 4. For each of males and females, there are 2 main effects of age and age$^2$. | Economic Burden of Asthma [25] |
| Asthma exacerbation | For a given year, sex, age, and the proportion of time spent in each of the control levels in that year (denoted $CT_j$ for $j = 1, 2, 3$), $N_i$, the number of exacerbations for individual $i$ labeled as having asthma is generated by a Poisson model:<br>$N_i \sim \text{Poisson}\left(\begin{array}{c} \mu\|CT_1, CT_2, CT_3, \\ \text{year, sex, age} \end{array}\right)$;<br>$\log\left(\mu\|CT_1, CT_2, CT_3, \text{ year, sex, age}\right) = \beta_0 + \sum_j \beta_j\, CT_j$,<br>where $\beta_0$ is determined by calibration as a function of year, sex, and age. | Economic Burden of Asthma [25], the GOAL study [26], Lee et al. [27], calibration |
| Exacerbation severity | There are four levels of exacerbation severity: 1 = mild, 2 = moderate, 3 = severe, and 4 = very severe. Given the number of exacerbations $N_i$ in a year and the past history of very severe exacerbations $n_{i4}^{\text{past}}$, the number of exacerbations $n_{ij}$ in each severity level $j$ is simulated for each individual $i$ by the Dirichlet-Multinomial distribution. A preliminary vector of probabilities for the severity levels is sampled from a Dirichlet distribution:<br>$\mathbf{w}_i^{pre} \sim \text{Dirichlet}\left(\boldsymbol{\alpha}\right)$.<br>If individual $i$ experienced any previous very severe exacerbations, their probability of very severe exacerbations is increased to yield:<br>$\mathbf{w}_i = \mathbf{w}^{pre}\left(n_{i4}^{\text{past}}\right)$ (see the main text for details).<br>The number of exacerbations by the severity levels is generated by a multinomial distribution:<br>$\left(n_{i1}, n_{i2}, n_{i3}, n_{i4}\right) \sim \text{Multinomial}\left(N_i, \mathbf{w}_i\right)$. | Yaghoubi et al. [28] Bateman et al. [29], Lee et al. [30] |
| **Payoffs** | | |
| Baseline utility | The utility values measured in quality-adjusted life years, ranging from 0 (death) to 1 (perfectly healthy) based on EQ-5D are used to assign the base utility of each individual by sex and age. | Yan et al. [31] |
| Disutility due to asthma control (per year) | The base utility value of individual $i$ labelled as having asthma decreases based on their asthma control status. Let $du_j^C$ denote the disutility due to having asthma control level $j$ in a year. Then the total disutility due to asthma control for individual $i$ in a year is given by:<br>$\sum_j du_j^C CT_j$. | McTaggart-Cowan et al. [32], Einarson et al. [33] |
| Disutility due to asthma exacerbation (per year) | The base utility value of individual $i$ labelled as having asthma decreases if they experience asthma exacerbations. Let $du_j^E$ denote the disutility due to experiencing a single asthma exacerbation of severity $j$ in a year and $n_j$ be the number of exacerbations of severity $j$ for $j$ = mild, moderate, severe, and very severe. Then the total disutility due to asthma exacerbations for individual $i$ in a year is given by:<br>$\sum_j du_j^E n_j$. | Lloyd et al. [34], Campbell et al. [35], Yaghoubi et al. [28] |
| Direct annual costs due to asthma control | Let $c_j^C$ be the direct annual costs due to asthma control level $j$ for individual $i$. Then the total cost due to asthma control for individual $i$ in a year is given by:<br>$\sum_j c_j^C CT_j$. | Yaghoubi et al. [28] |
| Direct annual costs due to asthma exacerbation | Let $c_j^E$ be the direct annual costs due to one asthma exacerbation of severity $j$ for individual $i$. Then the total cost due to asthma exacerbations for individual $i$ in a year is given by:<br>$\sum_j c_j^E n_j$. | Yaghoubi et al. [28] |

[a]The notation **1[*condition*]** is a univariate indicator with a value of 1 if the condition is met and 0 otherwise.

[b]For the initial version of the model. We are in process of incorporating additional risk factors.

[c]The notation $x * y$ means main effects of x and y as well as their interaction effects.

[d]The notation $\overrightarrow{\beta}$ means a vector of regression coefficients of the necessary dimension.

[e]The notation poly(x,q) means polynomials of x up to degree q (i.e., $x, x^2, \ldots, x^q$).

we used the Canadian Healthy Infant Longitudinal Development (CHILD) study, an ongoing representative birth cohort of 3,455 families [23]. We found that 29.3% (95% CI: 27.3%,31.1%) of the children in the study had parental asthma. We assumed that all individuals had this probability of having family history of asthma at birth, regardless of birth year, age, and sex, based on stable trends in asthma prevalence reported in national annual survey data from 2015 to 2022 [36]. To obtain an estimate of infant antibiotic exposure, we analysed infant antibiotic prescription data from the population-based administrative health databases of BC from January 2001 to December 2018 [37]. Then we fitted a negative binomial model to simulate the number of infant antibiotic prescriptions for each virtual individual (Supplementary Fig 1 in S1 File). Further details are provided in Supplementary Materials Section 2 in S1 File.

### 2.4. Asthma occurrence module

This module is responsible for assigning a label of asthma to simulated individuals. In consultation with the steering committee, we focused on labelled (i.e., diagnosed), rather than true, asthma states due to a lack of evidence on true underlying inflammation. Indeed, treatment strategies and disease management policies for asthma are mostly based on labelled asthma states.

For the initial population, the model uses a prevalence equation to determine whether an incoming virtual individual is labelled as having asthma in the base year of the microsimulation. For individuals not labelled as having asthma for each time cycle in the model, the model uses an incidence equation to determine whether they become labelled as having asthma. We assumed that immigrants have the same asthma incidence and prevalence rates as Canadians. Asthma was not modelled for children under 3 years of age, as it is difficult to perform and confirm asthma assessment for that age group [38]. Accordingly, the model does not assign any asthma label to children under 3 years of age. The equations for incidence and prevalence are provided in **Table 1**.

Values for the parameters in these equations were obtained in a stepwise fashion. We started with population-based age- and sex-specific 'crude' prevalence and incidence (Supplementary Fig 2 in S1 File). For the past (before 2000), we assumed the estimates of prevalence and incidence from 2000. For the future (2020 onwards), we assumed that current incidence and prevalence trends continued up to 2025 (referred to as the stabilisation year) and remained constant thereafter (Supplementary Figs 3 and 4 in S1 File). This assumption was based on the stability of recent trends in asthma prevalence [39]. Then we introduced asthma reassessment to calibrate the age- and sex-specific crude incidence and prevalence for each year. This mechanism is equivalent to remission, wherein an individual previously labelled with asthma can become non-asthmatic, return to the susceptible pool, and be at risk of asthma incidence in subsequent cycles. Next, we obtained the effect of two additional risk factors on asthma prevalence: family history of asthma at birth and exposure to antibiotics in the first year of life. For the former, we carried out an analysis using the CHILD study data; for the latter, we used the estimated effect based on a meta-analysis from another study [8]. Finally, we incorporated and calibrated those effects in both the asthma prevalence and incidence equations. Details are provided in Supplementary Materials Section 3 in S1 File.

### 2.5. Asthma outcomes module

Three main features of the disease course of asthma after diagnosis are asthma control, asthma severity, and exacerbations. Asthma control refers to how well asthma and the risk of adverse outcomes can be managed with risk factor modifications or treatment [40]. According to the definition by the Global Initiative for Asthma [40], at any given time, an individual with asthma is either uncontrolled, partially controlled, or well-controlled. Asthma severity refers to the innate intensity of the disease, conventionally classified into three levels: mild, moderate, and severe [40]. An asthma exacerbation is a sudden worsening of asthma symptoms, such as wheezing, coughing, and chest tightness, and is customarily classified into four levels: mild, moderate, severe, and very severe [40]. Exacerbation rate is inversely correlated with asthma control, as evidence suggests that achieving asthma control through reducing airway inflammation significantly reduces the risk of exacerbations [40,41].

**2.5.1. Asthma control.** To estimate the distribution of asthma control and severity in Canada, we used the data from the Economic Burden of Asthma (EBA) study [25]. EBA was a prospective representative observational study of 618 participants aged 1–85 years (74% are 18 years or older) with self-reported, physician-diagnosed asthma from BC. Asthma control and numbers of exacerbations were measured every 3 months for a year.

We fitted a model to this dataset for generating the proportion of time that an individual labelled as having asthma spends in each control level. This was a random-effects ordinal regression model with logit link and included age, age squared, sex, and their interactions as fixed-effects and individual as a random-effect [42]. For each individual labelled as having asthma, we sampled an individual-specific intercept from the estimated distribution of the random-effects as a proxy for asthma heterogeneity and severity, and with that intercept in the asthma control prediction model, we simulated the proportion of time spent in each of the control levels in each time cycle.

**2.5.2. Asthma exacerbation.** For each individual labelled as having asthma in each year, we used a Poisson regression model to simulate the number of exacerbations in each cycle conditional on the proportion of time spent in each of the asthma control levels by combining evidence from the EBA study [25] and the Gaining Optimal Asthma controL (GOAL) study [26]. Combining these two sources was deemed necessary because the EBA study alone did not have enough events for a robust estimation of exacerbation rate. The GOAL study was a one-year randomised, double-blind clinical trial with 3,421 individuals aged between 12 and 80 years with uncontrolled asthma at entry to the study, with asthma exacerbations as the primary outcome.

Combining evidence from the EBA study and the GOAL study, we obtained the annual exacerbation rate conditional on each asthma control level: rate(well-controlled) = 0.188, rate(partially-controlled) = 0.376, rate(uncontrolled) = 0.564. Then we simulated the total number of exacerbations in a year experienced by an individual with asthma as Poisson, where the log of the mean parameter is a linear function of the time that individual spent in each of the asthma control levels and a calibration intercept term. Details on exacerbations, including their severity, as well as on the calibration and initialisation of the exacerbation module, are provided in Supplementary Materials Section 4 in S1 File.

## 2.6. Payoffs module

The payoffs module is responsible for assigning health state utility values and costs. The utility values were measured in quality-adjusted life year weights (ranging from 0 [death] to 1 [perfectly healthy]) measured using EQ-5D-5L [43]. We obtained the baseline population average utility values for the Canadian population from a recent study [31]. As this study does not provide the utility values for less than 18 years of age, we assumed that the utility values started at 1 at birth and linearly interpolated to 18 years of age for each sex (where the utilities were 0.881 for females and 0.875 for males).

We modelled reduction in utility (disutility) values due to having asthma for each control level and for exacerbations (by their severity level). For the former, we used the data from a discrete choice experiment for 157 adult patients with asthma in Canada [32,33]. The disutility of well-controlled asthma was 0.06. This value was 0.09 for partially-controlled asthma and 0.10 for uncontrolled asthma. For the latter, we used the data from a prospective observational study that collected clinical and health-related quality of life information from 112 patients with moderate or severe asthma recruited at four asthma centres across the United Kingdom [28,34,35]. The disutility of having a mild exacerbation for a year was 0.32. This value was 0.44 for moderate exacerbation and 0.56 for very severe exacerbation. To estimate disutility for severe exacerbation, we took the midpoint between the disutility values of moderate and very severe exacerbations: 0.50. We assumed that a mild exacerbation lasted one week (leading to disutility of 0.32 * 1/52 = 0.006) and that exacerbations of other severity levels lasted two weeks, based on typical exacerbation duration reported in empirical studies [44].

Direct annual costs due to having asthma by the control levels and having exacerbation by the severity levels were extracted from Yaghoubi et al. [28] All costs were converted to 2023 CAD using historical inflation (e.g., 1 USD in September 2018 = 1.22 USD in September 2023 using the Consumer Price Index Calculator from the U.S. Bureau of Labor Statistics [https://www.bls.gov/data/inflation_calculator.htm]; 1 USD in 2023 = 1.36 CAD in 2023 using the Bank of Canada's

exchange rate [https://www.bankofcanada.ca/rates/exchange/annual-average-exchange-rates/]). To estimate the direct costs of having a severe exacerbation, we took the geometric mean of the direct costs of moderate and very severe exacerbations.

### 2.7. Implementation

The model is publicly available on the GitHub repository: https://github.com/tyhlee/LEAP.jl (Julia) and https://resplab.github.io/leap/ (Python). In Supplementary Materials Section 5 in S1 File, we describe how the structural equations in each of the five main modules were constructed and used to simulate individual-level characteristics and asthma events in the microsimulation (Supplementary Fig 5 in S1 File).

## 3. Results

We report the results of the face validity and internal validity assessments.

### 3.1. Face validity

The face validity assessments on structure, assumption, and parameters were performed iteratively in consultation with the steering committee throughout the model development. In the beginning of the model development, we observed that asthma prevalence would continue to increase and surpass observed values without modelling asthma remissions. As such, we incorporated the asthma reassessment to capture asthma remissions that often occur as children's immune systems and lungs become stronger. We checked whether the direction and magnitude of the effects of the risk factors were clinically sensible. Further, we checked whether asthma prevalence and outcomes changed in expected direction in response to changes in risk factors.

### 3.2. Internal validity

For the following internal validity assessments, we used results from one single simulation run with the entire Canadian population (approximately 38 million in 2019). Figures describing these assessments appear in Supplementary Materials Section 6 in S1 File.

**3.2.1. Demographics module.** The asthma model simulated mortality robustly as illustrated by close alignment of the simulated and estimated (prior to 2020) or projected (2020 onwards) values from Statistics Canada (Supplementary Fig 6 in S1 File). After incorporating the adjustment in mortality and net immigrants and emigrants, the model also generated the population growth and aging as estimated or predicted by Statistics Canada (**Fig 2**).

**3.2.2. Risk factors module.** The asthma model correctly produced the distribution of risk factors. The mean proportion of simulated individuals with family history of asthma was 29.3% (95% CI: 29.2,29.3) with the target value of 29.3%. The simulated trends in antibiotic exposure in the first year of life followed the observed trends well and appeared to plateau at 50 per 1,000 as programmed (**Fig 3**).

**3.2.3. Asthma occurrence module.** After incorporating calibration, the asthma model produced asthma prevalence rates in close alignment with target values (estimated or projected; **Fig 4**). Moreover, the differences between the target and simulated log ORs for asthma prevalence were just slightly larger than expected (median = 0.05; mean = 0.07), implying simulated values on average were modestly lower than the target values. The extreme differences (min = −0.68; max = 1.72) below the lower 1% quantile (−0.40) and above the upper 99% quantile (0.70) all occurred in combinations of risk factors with low probabilities.

**3.2.4. Asthma outcomes module.** The asthma model correctly produced the asthma control levels (Supplementary Fig 7 in S1 File). Asthma severity levels appeared to be correctly modelled (Supplementary Fig 8 in S1 File). The simulated rate of very severe exacerbations was slightly higher than the target value, as the risk of very severe

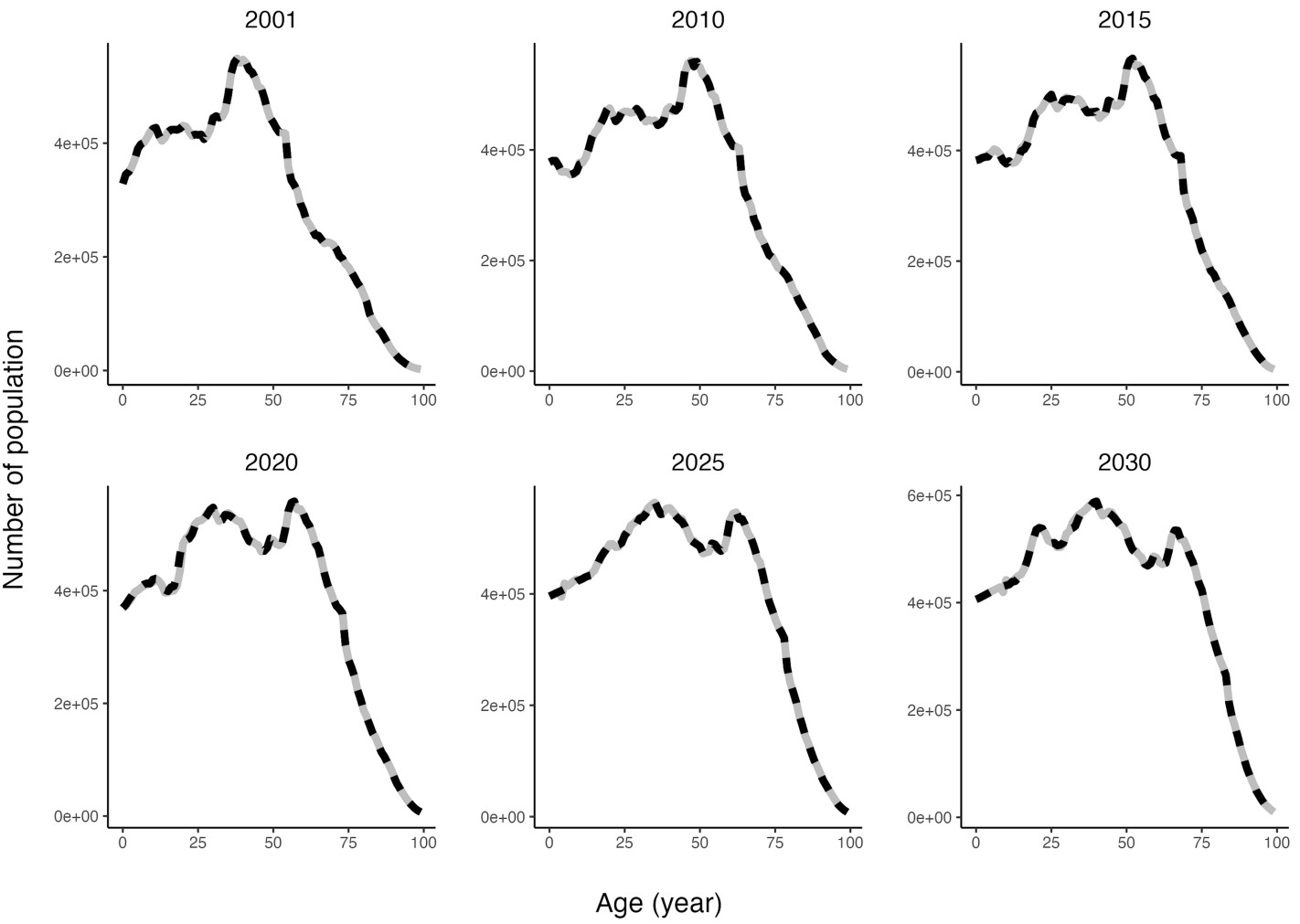

**Fig 2. Population by age across selected years from the model (grey solid) and from Statistics Canada (black dashed).**

exacerbations was increased for individuals with past history of very severe exacerbation (Supplementary Fig 9 in S1 File). However, differences in the ratio of target and simulated very severe annual exacerbation rates on the log scale were modest, ranging from −0.15 to 0.04 with median and mean values of −0.07 and −0.08.

   **3.2.5. Payoffs module.** Simulated direct annual costs increased over time as the number of asthma patients increased (Supplementary Fig 10 in S1 File). However, the per-patient costs due to exacerbations decreased, as expected, due to decreasing rates of severe and very severe exacerbations. In contrast, the per-patient costs due to managing asthma control were correctly stable over the years as assumed in the simulation model. The total direct annual costs were consistent with the projected values from an independent study from a non-profit Canadian research institution (e.g., 2.3 billion vs. 2.4 billion in 2030) [45]. Finally, simulated utility values closely matched the target values with a root mean squared error of 0.01 (**Fig 5**).

## 4. Discussion

We established key components for a reference policy model for asthma in Canada with a focus on early childhood asthma and on primary prevention strategies. To populate this model, we used multiple data sources and evidence

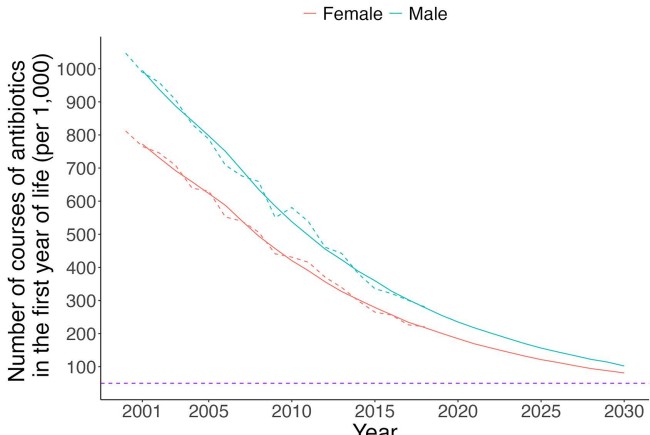

**Fig 3. Rate of infant antibiotic prescriptions by sex (red: females; blue: males) for simulated (solid) and target (dotted) values from the model and the population-based administrative database, respectively, with the floor rate of 50 per 1,000 (purple).**

synthesis methods to obtain the highest quality of available evidence and to maximise generalisability. This included secondary analyses of several population-based and clinical datasets to estimate disease prevalence, variations in control levels, and exacerbation rate and severity.

We performed several face validity and internal validity assessments to ensure the model is properly structured and correctly implemented such that its outputs match the expected patterns. LEAP performed satisfactorily in the internal validation studies. With calibrated mortality and net immigrants and emigrants, the model replicated the population growth quite well. Of note, Statistics Canada has recently released detailed migration data, which will allow for the modelling of actual numbers in the next version [46]. The marginal asthma prevalence was replicated well up to 2030. However, we observed that the simulated values were slightly underestimated among young adults thereafter due to the lack of evidence on asthma reassessment prompting reversals in diagnosis. One could overcome this limitation by modelling true asthma states and introducing an asthma diagnosis module as a means to calibration. Calibration of asthma prevalence by the risk factors was satisfactory in general. For better calibration, one could allow a more flexible form of the risk factor equations for asthma incidence. Moreover, we observed a slight overestimation of severe exacerbation rates, particularly among males. To minimise these discrepancies, more granular calibration techniques could be employed, such as the use of deep learning based Bayesian calibration [47].

The LEAP model was designed to provide a unified framework under which different interventions can be realistically and consistently compared. This objective motivated several design choices. First, we used an open-population structure that enables modelling of realistic aspects of population aging and adopting health interventions, such as gradual and imperfect market penetration. Further, we used an individual-level simulation platform to robustly model the vast parameter space created by the many combinations of patient and disease characteristics and event histories. In addition, major focus was placed on designing an easily accessible, transparent, and expandable platform.

The limitations of this work should be recognised. Extrapolating beyond available data inherently introduces uncertainty. For instance, the asthma incidence and prevalence equations in the LEAP model were derived from the BC administrative databases but were applied to the entire Canadian population. This choice was justified by the minimal difference in asthma prevalence between BC and Canada, as estimated from the Canadian Community Health Survey data. Nevertheless, relying on a single province may not fully capture regional heterogeneity in clinical practices or disease trajectories, potentially limiting geographic generalisability of the model to other provinces or territories. It would be more desirable to estimate incidence and prevalence rates from a sample representative of the entire country or explicitly differentiate

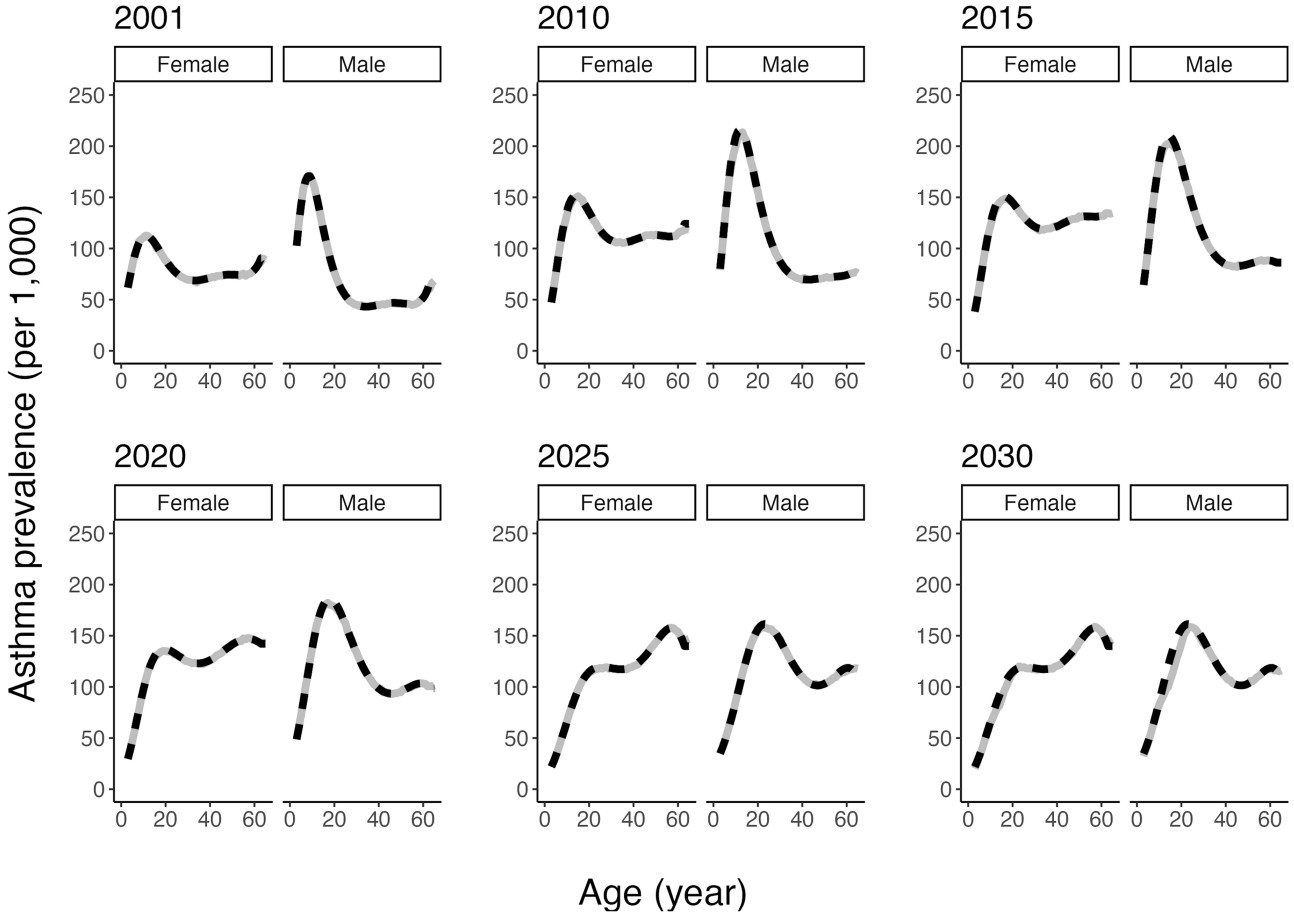

**Fig 4. Asthma prevalence rates per 1,000 general population by sex (left: females; right: males) from the model (grey solid) and estimated or projected by Statistics Canada (black dashed).**

asthma occurrence and outcomes between provinces and territories [48,49]. Based on the stable trends in asthma prevalence reported in national annual surveys from 2015 to 2022 [36], we assumed a constant prevalence for family history of asthma across all simulated individuals. However, as some Canadian provinces have reported increasing trends in asthma prevalence [50], this assumption may lead to a misestimation of family history prevalence in those regions. Future work should incorporate regional heterogeneity and account for potential cohort effects on the parameters.

We calibrated severe exacerbation rates against observed values in the national Canadian hospitalisation database. However, this calibration does not guarantee that the rates for other severity levels align with current practice. This discrepancy arises because the baseline distribution of severity levels was derived from cohort studies, which may not fully represent the Canadian population or reflect shifts in treatment paradigms, such as widespread use of biologics.

Furthermore, we extrapolated health state utility values for children under 18 years of age under the assumption of perfect health at birth. While this interpolation matched the utility values reported for children aged 6 years or older in other Canadian studies using the Health Utility Index Mark 3 [51], it may not well represent the utility values for those under 6 years of age, as the assumption of perfect health at birth may be considered overly optimistic. We suggest sensitivity analyses utilising lower utility values for this younger age group. In addition, we could not incorporate caregiver or family utility effects due to lack of evidence in Canada. The current version of the LEAP model did not incorporate indirect costs, such

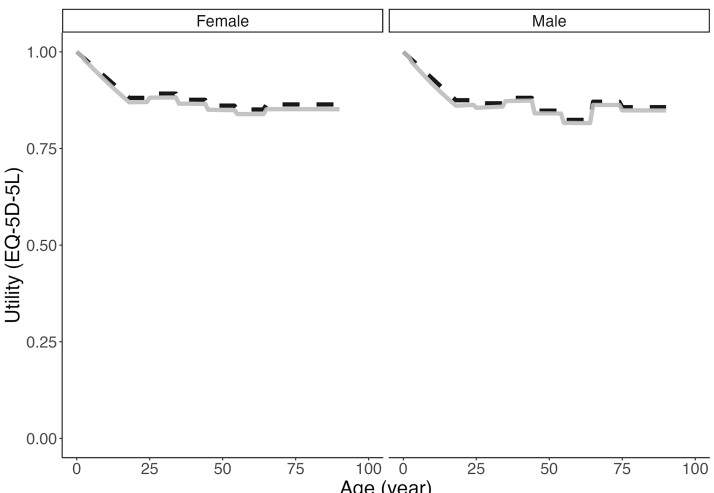

**Fig 5. Comparison of simulated (grey solid) and target (black dashed) utility values by sex (left: females; right: males) across age.**

as productivity losses from missed school days for children and work absenteeism for adults, which represent a substantial proportion of the economic burden of asthma [52,53].

Currently, the natural course of asthma is modelled through labelled/diagnosed, rather than true, asthma state based on objective measures of reversible airflow obstruction and airway hyperresponsiveness. This was deemed sufficient for the first set of policy questions LEAP is intended to tackle, which are about preventive strategies that reduce the risk of future asthma. The outcomes of interest related to such policies are future asthma cases and their related outcomes. To address some other policy questions, such as the benefit of screening, case detection, and case verification policies, true asthma states should be modelled in the future upon accumulation of sufficient evidence on true asthma states. Similarly, we currently incorporated a selective set of asthma risk factors, but many more risk factors, such as exposure to air pollution, need to be developed to comprehensively capture the disease course of asthma.

The current version of the asthma model does not account for uncertainty in parameters [54]. Our conscious choice was to initially emphasise proper modelling of the structure. Future work requires specification of the parameter distributions and updating calibration procedures that are specifically tuned for probabilistic projections [55]. Further, future efforts should focus on dedicated external validation studies to assess the generalisability of the LEAP model across different populations and time periods. This includes temporal validation to ensure the model structure remains robust across time as well as to geographic validation through strategic collaborations both locally (e.g., using the pan-Canadian respiratory electronic health records [56]) and internationally (e.g., via the Copenhagen Prospective Studies on Asthma in Childhood [57] and the International Severe Asthma Registry [58]). In the impact analysis study using the LEAP model [8], the asthma occurrence module was externally validated against a 'land-mark' study. Additional components of the model need to be externally validated to increase its credibility among stakeholders, and each application study using the LEAP model should be accompanied by validation against relevant landmark studies.

There are numerous existing and upcoming programs, policies, and interventions whose population-level impact on asthma and their value proposition can be explored using the LEAP model. As a first use case, the LEAP model has been used to evaluate the impact of reducing unnecessary use of antibiotics among infants on the burden of asthma [8]. Beyond obvious cost-savings from prescribing fewer antibiotics, this study has shown that a declining trend of antibiotic prescriptions in BC prevented 10,053 asthma incident cases from 2001 to 2018, compared to a counterfactual scenario in which the prescription rate remained at the 2001 value. To identify priority areas, we have held a series of discussion

sessions with the 8 stakeholders representing clinical (pediatric and adult respirologist), government (Health Canada), non-profit (BC Lung Foundation), and patient perspectives [59]. We are actively expanding the LEAP model (e.g., incorporating additional risk factors such as smoking and outdoor air pollution exposure and indirect costs) and plan to address the following priority areas with buy-in from stakeholders: outdoor air pollution and cost-effectiveness of a portable HEPA air cleaner distribution program, expanded access to smoking cessation programs, and respiratory syncytial virus vaccine for prevention of early childhood infection. The model platform will be governed by an advisory board (consisting of the steering committee and stakeholders) that oversees the model update mechanism, stakeholder engagement, and validation to ensure long-term maintenance, adaptation, and policy relevance (https://leap.core.ubc.ca).

## 5. Conclusions

Projecting the outcomes of decisions is a prerequisite for evidence-informed decision making. The LEAP model addresses the concerns with existing asthma models on generalisability, transparency, accessibility, and granularity in details [14,15]. It represents consolidated efforts in evidence generation, evidence synthesis, and validation. With further development and external validation planned, the value of this asthma reference policy model will continue to be enhanced, providing an invaluable platform for focusing resources to inform optimal investment in asthma prevention and care.

## Supporting information

**S1 File. Supplementary materials.**
(DOCX)

## Author contributions

**Conceptualization:** Tae Yoon Lee, John Petkau, Mohsen Sadatsafavi.

**Data curation:** Tae Yoon Lee, Kate M. Johnson, Stuart E. Turvey, Padmaja Subbarao, Mohsen Sadatsafavi.

**Formal analysis:** Tae Yoon Lee.

**Funding acquisition:** Mohsen Sadatsafavi.

**Investigation:** Tae Yoon Lee, John Petkau.

**Methodology:** Tae Yoon Lee, John Petkau, Mohsen Sadatsafavi.

**Software:** Tae Yoon Lee.

**Supervision:** John Petkau, Mohsen Sadatsafavi.

**Validation:** Tae Yoon Lee.

**Visualization:** Tae Yoon Lee.

**Writing – original draft:** Tae Yoon Lee.

**Writing – review & editing:** Tae Yoon Lee, John Petkau, Kate M. Johnson, Stuart E. Turvey, Amin Adibi, Padmaja Subbarao, Ainsleigh Hill, Mark Ewert, Mohsen Sadatsafavi.

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
