## [Decision Letter · Decision Letter 0]

9 Jan 2026

PONE-D-25-58120

Development of an asthma policy model for Canada: Lifetime Exposures and Asthma outcomes Projection

PLOS One

Dear Dr. Lee,

Thank you for submitting your manuscript to PLOS ONE. After careful consideration, we feel that it has merit but does not fully meet PLOS ONE’s publication criteria as it currently stands. Therefore, we invite you to submit a revised version of the manuscript that addresses the points raised during the review process.

We look forward to receiving your revised manuscript.

Kind regards,

Tsai-Ching Hsu, Ph.D.

Academic Editor

PLOS One

Journal Requirements:

Reviewers' comments:

Reviewer's Responses to Questions

**Comments to the Author**

1. Is the manuscript technically sound, and do the data support the conclusions?

Reviewer #1: Yes

Reviewer #2: Yes

Reviewer #3: Yes

2. Has the statistical analysis been performed appropriately and rigorously?

Reviewer #1: No

Reviewer #2: Yes

Reviewer #3: Yes

3. Have the authors made all data underlying the findings in their manuscript fully available?

Reviewer #1: Yes

Reviewer #2: Yes

Reviewer #3: No

4. Is the manuscript presented in an intelligible fashion and written in standard English?

Reviewer #1: Yes

Reviewer #2: Yes

Reviewer #3: Yes

5. Review Comments to the Author

Reviewer #1: 1. Lines 419–421: The authors state, “The current version of the asthma model does not account for uncertainty in parameters... Our conscious choice was to initially emphasise proper modelling of the structure.” Can the authors justify publishing a policy model that cannot currently generate confidence intervals or credible intervals for its projections? When will the parameter distributions be specified?

2. Lines 201–204: The model simulates “labelled (diagnosed)” asthma rather than “true” physiological asthma. How does the model differentiate between an intervention that reduces disease onset versus an intervention that simply reduces diagnosis rates? If true asthma states are not modeled, how can the model evaluate screening or case-detection policies effectively?

3. Lines 184–186: The model currently only includes Age, Sex, Family History, and Antibiotic Exposure. Given the stated objective to support “health policy planning” (Line 31), is the inclusion of only one modifiable risk factor (antibiotics) sufficient to validate the model structure? Why were strong predictors like prematurity or maternal smoking excluded from the first version?

4. Lines 400–403: The authors extrapolate British Columbia (BC) incidence/prevalence data to the entire Canadian population, justifying it by minimal difference in CCHS data. Canada has significant regional heterogeneity in asthma rates (e.g., Atlantic Canada often shows higher rates than the West). Relying on BC administrative data (which depends on BC-specific billing codes and physician behaviours) risks biasing the national model. Did the authors perform a sensitivity analysis using provincial adjusters based on CCHS data to see if regional variations alter the aggregate national outcomes?

5. Line 140: The model uses an annual time cycle. Asthma exacerbations are acute events lasting days. Infants undergo rapid developmental changes. Is an annual cycle granular enough to capture the immediate effects of infant antibiotic exposure (which occurs in months 0-12) or the seasonality of exacerbations?

6. Line 191: The model assumes all individuals have a 29.3% probability of family history, regardless of birth year. Since asthma prevalence has increased over the last 30 years (according to the Introduction), shouldn't the rate of parental history also be increasing for later cohorts? A child born in 2020 is more likely to have asthmatic parents than a child born in 1980.

7. Line 220: The authors introduced asthma reassessment to calibrate prevalence. Is this mechanism structurally equivalent to remission? Does an individual who loses the asthma label return to the “susceptible” pool and become at risk for ‘incidence’ again, or are they immune? The handling of ‘re-diagnosis’ is crucial for longitudinal accuracy.

8. Line 266: The rates (0.188 for WC, 0.564 for UC) seem to be derived from the GOAL study (2004) and EBA study. Treatment paradigms have shifted significantly since 2004 (e.g., widespread use of biologicals, SMART therapy). Are these exacerbation rates representative of current Canadian clinical practice?

9. Line 278: The model assumes utility starts at 1 (perfect health) at birth. This assumption is contrary to most HTA guidelines, as infants often have temporary health states (colic, infections) that reduce utility below 1.0. Does this assumption skew QALY calculations for early-childhood interventions?

10. Line 354: ‘The simulated rate of very severe exacerbations was slightly higher than the target value.’ How does this overestimation impact the cost analysis, given that very severe exacerbations (hospitalizations) are likely the primary cost drivers?

11. Line 115: “June 31, 2025”. June has only 30 days.

12. Line 164 (Table 1 Footnote): Please ensure mathematical notation is consistent throughout the table and text.

13. Line 210: “Asthma was not modelled for children under 3 years of age”. How does this interact with the ‘Infant Antibiotic Exposure’ module? If exposure happens at age <1, but asthma "starts" at age 3, is the risk factor stored as an attribute until the agent turns 3?

14. Line 296 (Inflation): “1 USD in 2018 = 1.22 USD in 2023”. This implies a cumulative inflation of 22% over 5 years. While inflation was high recently, please cite the specific CPI calculator or source used for this conversion.

AFTER A DEEP REVIEW I FOUND SOME QUESTIONS MENTIONED HERE ABOVE AND MISTAKES AND ERRORS. I request to take care of these.

Reviewer #2: I am pleased to have the opportunity to review this manuscript. This study presents the methodology of the LEAP model, developed to support asthma prevention and policy-oriented decision-making, and represents an important contribution as a national-level, open-source policy model. Overall, the manuscript is well designed and presents an interesting and timely study.

1) The LEAP model is designed and calibrated for a specific population—namely, the Canadian population—within the context of a particular healthcare system. As such, conducting external validation in the conventional sense may be inherently challenging due to these structural constraints. Nevertheless, to enhance the generalizability and credibility of the model, a more detailed discussion of potential strategies for external validation—such as validation across different regions, time periods, or independent data sources—would be valuable in future work.

2) At present, the model incorporates a relatively limited set of risk factors, primarily focusing on basic demographic characteristics and selected early-life exposures. Nevertheless, given the stated goal of supporting population-level policy evaluation, further incorporation of additional environmental, behavioral, and clinical risk factors would substantially enhance the model’s applicability and policy relevance.

3) The selection of risk factors appears to be grounded in a previously published concept map developed through a modified Delphi process. However, the detailed procedures of this consensus-building process are not fully described in the current manuscript. Given that this step is foundational to the construction of the model, greater transparency regarding the Delphi process is critical. In addition, it would be helpful to clarify whether future updates and expansions of the model will follow the same methodological framework.

4) Many parameters used in the long-term projections appear to be derived from short- to medium-term data and are treated as time-invariant. Incorporating parameter uncertainty, particularly regarding the temporal stability of these estimates, would strengthen the robustness of the model’s long-term projections.

5) As a minor point, the resolution of several figures appears to be low. Providing higher-resolution figures would improve the clarity and readability of the manuscript.

Reviewer #3: Thank you for this important modelling work. My comments are minor:

Abstract

- Under results, it was stated that "Age-sex stratified demographic projections from the model closely matched projections". Does the second “projection” refer to those published from national surveys?

Methods

- May I suggest putting key equations such as Equation 2 from the supplementary material back to the main text?

- Line 190: “We found that 29.3% (95% CI: 27.3%,31.1%) of the children in the study had parental asthma.” Were these estimates derived by the authors from the cited study?

- Is there a reason why published transition probabilities were not used to model the progression of asthma across the control states?

Result

- Line 363 - Does the decline in severe and very severe exacerbations overtime algin with expectation?

Discussion

- Line 405 – what does XXX represent?

- Is the inclusion of indirect cost into LEAP a potential direction of future work?

Supplementary Material

- May I suggest adding a table of content with links to the different sections/pages.

- Page 9 of supplementary material - in line 2, "crude asthma and incidence"

- Since direct cost is an outcome of interest, could the authors detail its components and corresponding estimates?

6. PLOS authors have the option to publish the peer review history of their article (what does this mean?). If published, this will include your full peer review and any attached files.

Reviewer #1: **Yes:** sri chakradhar challagali

Reviewer #2: No

Reviewer #3: No

---

## [Decision Letter · Decision Letter 1]

22 Apr 2026

Development of an asthma policy model for Canada: Lifetime Exposures and Asthma outcomes Projection

PONE-D-25-58120R1

Dear Dr. Lee,

We’re pleased to inform you that your manuscript has been judged scientifically suitable for publication and will be formally accepted for publication once it meets all outstanding technical requirements.

Kind regards,

Tsai-Ching Hsu, Ph.D.

Academic Editor

PLOS One

Additional Editor Comments (optional):

All comments have been addressed in accordance with the reviewers' suggestions.

Reviewers' comments:

Reviewer's Responses to Questions

**Comments to the Author**

1. If the authors have adequately addressed your comments raised in a previous round of review and you feel that this manuscript is now acceptable for publication, you may indicate that here to bypass the “Comments to the Author” section, enter your conflict of interest statement in the “Confidential to Editor” section, and submit your "Accept" recommendation.

Reviewer #1: All comments have been addressed

Reviewer #4: All comments have been addressed

2. Is the manuscript technically sound, and do the data support the conclusions?

Reviewer #1: Yes

Reviewer #4: Yes

3. Has the statistical analysis been performed appropriately and rigorously?

Reviewer #1: Yes

Reviewer #4: Yes

4. Have the authors made all data underlying the findings in their manuscript fully available?

Reviewer #1: Yes

Reviewer #4: Yes

5. Is the manuscript presented in an intelligible fashion and written in standard English?

Reviewer #1: Yes

Reviewer #4: Yes

6. Review Comments to the Author

Reviewer #1: I recommend that this manuscript be Accepted for publication in PLOS ONE following a final check of the conclusion for balanced reporting of the newly added limitations. The LEAP model is a non-trivial and valuable contribution to the field of respiratory health and clinical engineering integration in Canada. The authors' responses have satisfied my primary concerns regarding the model's structural logic and factual accuracy.

Reviewer #4: The revised manuscript demonstrates substantial improvement and has addressed the concerns raised in earlier rounds. The methodology is clearly explained, the data presentation is consistent, and the discussion now aligns well with the stated objectives. Ethical standards and publication guidelines appear to be followed, and I see no issues with dual publication. Overall, the article is well‑structured, contributes meaningfully to the field, and is suitable for publication. I recommend acceptance.

7. PLOS authors have the option to publish the peer review history of their article (what does this mean?). If published, this will include your full peer review and any attached files.

Reviewer #1: **Yes:** sri chakradhar challagali

Reviewer #4: **Yes:** Sura Saad Abdullah

---

## [Editor Report · Acceptance letter]

PONE-D-25-58120R1

PLOS One

Dear Dr. Lee,

I'm pleased to inform you that your manuscript has been deemed suitable for publication in PLOS One. Congratulations! Your manuscript is now being handed over to our production team.

Kind regards,

on behalf of

Dr. Tsai-Ching Hsu

Academic Editor

PLOS One